# Wen Dan Tang: A Potential Jing Fang Decoction for Headache Disorders?

**DOI:** 10.3390/medicines9030022

**Published:** 2022-03-04

**Authors:** Saroj K. Pradhan, Yiming Li, Andreas R. Gantenbein, Felix Angst, Susanne Lehmann, Hamdy Shaban

**Affiliations:** 1Research Department Rehaklinik, TCM Ming Dao, ZURZACH Care, 5330 Bad Zurzach, Switzerland; y.li@tcmmingdao.ch; 2Research Department, Swiss TCM Academy, 5330 Bad Zurzach, Switzerland; 3Research Department, Nanjing University of Chinese Medicine, Nanjing 210029, China; 4Neurology & Neurorehabilitation Department Rehaklinik, ZURZACH Care, 5330 Bad Zurzach, Switzerland; andreas.gantenbein@zurzachcare.ch; 5Research Department Rehaklinik, ZURZACH Care, 5330 Bad Zurzach, Switzerland; felix.angst@zurzachcare.ch (F.A.); susanne.lehmann@zurzachcare.ch (S.L.); 6Department of Private Psychiatry Clinic of UPK, University Psychiatric Clinics, 4002 Basel, Switzerland; hamdy.shaban@upk.ch

**Keywords:** Chinese herbal medicine, Wen Dan Tang, traditional Chinese medicine, headache disorders, migraine, tension-type headache

## Abstract

Background: Chinese herbal medicine is considered relatively safe, inexpensive, and easily accessible. Wen Dan Tang (WDT), a Jing Fang ancient classical Chinese herbal formula with a broad indication profile has been used for several centuries in China to treat various illnesses. Question: Are there evidence-based clinical trials that show that WDT has a significant impact on the treatment of various diseases, especially in patients with migraine and tension-type headaches (TTH)? Methods: This study is based on an online database search using PubMed, Medline, Cochrane Library, AcuTrials, Embase, Semantic Scholar, Jstor, internet research, and review of ancient and modern Chinese medical textbooks regarding WDT and its compounds. Results: There were no studies on WDT in migraine and TTH; therefore, this work gathers and describes data for every single compound in the formula. Conclusion: This study suggests that the bioactive compounds found in WDT composition show potential in treating patients with neurological, psychiatric disorders, cardiovascular diseases, metabolic syndrome, and digestive disorders. Some coherence between WDT in headache reduction and improvements in the quality of life in patients with migraines and TTH could be evaluated, showing positive results of WDT in these patients.

## 1. Introduction

The implementation of Traditional Chinese Medicine (TCM) for the prevention and treatment of numerous medical conditions has lasted for thousands of years, predominantly in China and Asia, but also in the Western world in the last few decades.

Wen Dan Tang (WDT), also known as “Warm the Gallbladder decoction”, is a classical famous Chinese herbal formula containing eight components derived from the book Treatise on Three Categories of Pathogenic Factors: *Pinellia ternata* (Thunb.) Makino (Ban Xia), *Bambusa tuldoides Munro* (Zhu Ru), *Citrus aurantium* L. (Zhi Shi), *Citrus reticulata Blanco* (Chen Pi), *Glycyrrhiza uralensis* Fisch (Gan Cao), *Poriae sclerotium* cocos albae (Fu Ling), *Zingiber officinale* Roscoe (Sheng Jiang), and *Ziziphus jujuba* Mil (Da Zao) (Table 1) [1,2]. 

Classically, WDT is a TCM prescription for disorders of the spirit and has been used for 100 years to treat symptoms such as schizophrenia. Deng and Xu have shown that WDT may have some positive short-term antipsychotic effects compared to placebo or no treatment. Nevertheless, combining WDT with an antipsychotic reduced the adverse effects of antipsychotics [19]. Clinically, WDT is used to treat psychiatric disorders (such as schizophrenia, major depressive disorder, and anxiety), insomnia, stroke, digestive disorder, metabolic syndrome, and cardiovascular disease [19,20,21,22,23,24,25,26,27,28,29]. The application of WDT in various syndromes is illustrated in Table 2. Additionally, WDT is considered relatively safe, economical, and obtainable [19].

Sun Si Miao (580–682 A.D.), who was honored as the “King of Herbal Medicine”, completed a 30-volume encyclopedia entitled Essential Recipes for Emergent Use Worth A Thousand Gold, including 4500 Chinese herbal formulae and a treatise on medical practice [33]. He describes in the mentioned encyclopedia, among other things, the components, properties, function, and preparation of WDT [34]. The classification method of disease disorders in this compendium derives from the Yellow Emperor’s Inner Canon, Treatise on Cold Pathogenic and Miscellaneous Diseases, and General Treatise on Causes and Manifestations of All Diseases [35].

In 2018, Sun Si Miao’s WDT was listed in the Catalogue of Ancient Classics Formulae (ACF) among the other 99 ACFs by the National Administration of Traditional Chinese Medicine [1]. 

WDT is available as a classical Chinese decoction, granules, pills, or hydrophilic concentrate. This hydrophilic concentrate from the Dr. Noyer AG/TCM pharmacy in Switzerland is produced according to the Kumagawa method. Each substance is preserved for one night in purified water and is then extracted with the so-called Kumagawa-Extractor. During this process, the substances are dissolved as in a decoction process. At the end of the procedure, glycerin is added to ensure the durability of the hydrophilic concentrate. Only boiled water as a solvent is used in the Kumagawa-Extractor. The extract obtained is therefore very similar to decoction but contains more API agents. This means that smaller quantities can be administered for an identical result compared to decoction [36]. 

Traditionally, the compounds of WDT are immersed in 1200 mL water for 30 min and then boiled in a decoction pot. After the boiling process, the decoction simmers for 30 to 40 min. The extract is filtered, and usually the dosage of 300 mL/day is administrated for drinking. Zhang et al. stated in their study that the extraction of WDT applying the UHPLC-QQQ-MS/MS method was more beneficial than the traditional extraction technique with water because the boiling process negatively affects the stability of the molecule gingerol. In addition, instead of methanol, acetonitrile was utilized for the chromatography, and therefore, better isolation of the individual components could be achieved. The complete extraction method could be performed within 30 min in the lab. This procedure is thus not more time-consuming than the traditional method, and more pure gingerol extracts can be obtained with it [2].

Headache disorders (HD) affect all ages and genders and have become a widespread public health burden [37]. Headache not only influences the individuals’ constricted quality of life (QoL) and health but also entails enormous economic costs [38].

The present valid International Classification of Headache Disorders (ICHD-3) published in 2018 [39] distinguishes between primary and secondary headaches, as well as neuropathies and facial pain, with more than 300 different types of headache, presented in a hierarchical structure for diagnosis. The most prevalent HD is migraine and tension-type headache (TTH). 

According to TCM, a person is healthy when Yin and Yang, vital energy (Qi), and blood in the body are in balance and the Qi can flow freely in the conduit pathways. All complaints arise by disorders aroused by blockages of Qi and/or blood or lack of Qi and/or blood and when Yin and Yang in the body become unbalanced [40]. 

Endogenous factors for headaches from the point of TCM’s view are mostly hyperactivity of liver Yang, deficiency of Qi and blood, spleen, kidney weakness, and ascending stagnated fire to the brain [41].

The Western medicine pharmacotherapy treatment for HD includes acute and preventive medications [42]. In comparison to Western medicine, the treatment with WDT can hypothetically be applied for all types of HD.

Additionally, there have been numerous research studies showing WDT usage in the treatment of various diseases that might lead to HD [19,20,21,23,24,25,26,27,29]. 

Thus, WDT might be beneficial in the prophylactic treatment for migraine with few adverse effects. The most common adverse events reported by Huang et al. in their meta-analysis were loose stool, dizziness, poor appetite, fatigue, and xerostomia. However, the observed AEs were mild [25]. 

From our clinical experience over the years, a very large number of patients reported an improvement in their HD and their QoL after WDT intake. Therefore, it was our approach to review the literature regarding the application of WDT in migraine and TTH. In our review, we questioned if there are evidence-based clinical trials that show that WDT has a significant impact on the treatment of HD, especially migraine and TTH.

## 2. Methods

### 2.1. Search Strategy

Research data were acquired from PubMed, Medline, Cochrane Library, AcuTrials, Embase, Semantic Scholar, Jstor, and the internet; additionally, ancient and modern Chinese medical textbooks regarding WDT were reviewed, and studies based on our aim were selected.

Search terms were as follows: headache disorders, Wen Dan Tang, Wen Dan Tang pharmacological reaction, migraine, tension-type headache, Wen Dan Decoction, WDT mechanisms of action.

### 2.2. Inclusion Criteria

Inclusion criteria contained reviews and peer-reviewed research articles exploring the effects of WDT. 

## 3. Results

Systematic reviews and peer-reviewed research articles were evaluated for this study. The results were analyzed, classified, and summarized. There were no studies on WDT in migraine and TTH; therefore, results and studies for the single compounds are described instead.

## 4. Discussion

In ancient China, WDT was applied to arouse courage in a person. According to TCM, WDT’s function is to remove heat-phlegm and harmonize the gallbladder and the stomach. The effects of WDT are sedative, anxiolytic, anti-depressive, regulate the flow of vital energy (Qi), expectorant, heat-eliminating, harmonize the stomach, and encouragement [43,44]. As the gallbladder is the source of courage in TCM, the phlegm heat may cause anxiety, restlessness, insomnia, and agitation. The flow of the rebellious Qi in the stomach may cause stomach upset. WDT clears phlegm and heat, regulates Qi, harmonizes the gallbladder and the stomach, and calms the spirit [45]. 

Modern scientific research provided evidence that WDT contains compounds including flavonoids, phenols, alkaloids, triterpenoid organic acid, polysaccharides, and phosphodiesterase inhibitors (Table 3), which have been found to possess neuroprotective, neuromodulation, anti-mutagenic, antioxidant, antiemetic, antithrombotic, antipyretic, and anti-inflammatory effects [2,30,46,47,48,49,50,51,52]. 

### 4.1. Alkaloids

Alkaloids are biologically active, organic, nitrogen-containing compounds that widely occur in various plant families such as the nightshade, poppy, and buttercup families [54]. The nitrogen atom(s) of true alkaloids is usually within a heterocyclic ring and derives from an amino acid. Alkaloids are often optically active, mostly left-handed, and, in the pure form, normally colorless crystalline substances [55].

The bioactive alkaloid within WDT is synephrine [2,30], which is a phenylethylamine derivative [56], also found in bitter orange (*Citrus aurantium* L. from the Rutaceae family) [57]. There are three isomeric forms of synephrine: m-synephrine, p-synephrine, and o-synephrine. P-synephrine may act on alpha-1 and beta-3 adrenoreceptors [58]. It causes a stimulation through its direct binding to the alpha-adrenoreceptor, which leads to a contraction of the smooth muscles, thus causing vasoconstriction of the blood vessels, e.g., in the mucous membranes [59]. The rhinitis-induced secondary headache can, for instance, be reduced by the bioactive substance synephrine [60].

Synephrine belongs to the trace amines synthesized from aromatic amino acids in mammals [61]. An increased level of plasma trace amines was shown to occur in both cluster headache and migraine patients. The alteration in biogenic amine metabolism is one of the characteristics of primary headache sufferers [62]. Additionally, dopamine and trace amines’ abnormal levels may contribute to the metabolic cascades that predispose headaches’ occurrence [63]. 

Migraine is a neurovascular disorder related to the impairment of the cerebral nerves and blood vessels. Calcitonin gene-related peptide (CGRP) is the most effective peptidergic vasodilator of peripheral and cerebral blood vessels. CGRP is unleashed to sensory nerves during severe migraine outbreaks and, for a long time, has been considered to play a very important role in the pathophysiology of migraine. Farooqui T. reported a hypothetical molecular mechanism underlying cluster and/or migraine headaches. One of the most accepted mechanisms indicates that the release of CGRP results in vasodilation and cranial meningeal stimulation, causing primary headache [64]. As synephrine has been described to exert vasoconstrictive effects, it can be assumed that the consumption of synephrine could contribute to the reduction in the so-caused primary headaches. 

The beta-3-adrenoreceptors are located in white and brown adipocytes [65] and play an important role in lipolysis and thermogenesis [59]. Synephrine activates the beta-adrenoreceptor and elevates cyclic AMP (cAMP) levels due to adenylyl cyclase activation. The rise in cAMP levels, which might lead to migraine [66], leads to a surge in the release of fatty acids from adipose tissue and consequently promotes fat oxidation by increasing thermogenesis [67]. In a study published in the British Journal of Clinical Pharmacology by Gutiérrez-Hellín et al., p-synephrine showed a significant increase in fat burning at low-to-moderate exercise intensity without changes in heart rate or blood pressure [68,69,70].

P-synephrine is chemically similar to ephedrine, but in contrast to synephrine, ephedrine exhibits bindings to beta-1, beta-2, and beta-3 receptors, leading to more adverse effects such as hypertension, tachycardic arrhythmias, hyperthyroidism, and an increase in respiratory rate [71]. Consequently, p-synephrine cannot be categorized as a stimulant [72,73].

P-synephrine consumption at recommended levels has been shown to be safe [57]. The low toxicity of p-synephrine and *Citrus aurantium* L. extracts in mice tested in an in vitro model at a high dose was shown by Arbo et al. and Rossato et al. to cause low or insignificant cardiotoxicity [74,75,76,77]. 

In a randomized controlled trial, Bond et al. compared the effect on headache frequency in women with comorbid migraine and overweight/obesity and concluded that weight loss would improve the QoL of the affected person and might be promising in reducing migraine headaches’ frequency [78].

Kaats et al. provided evidence in a randomized, double-blinded, placebo-controlled study that a chocolate-flavored chew with a bitter orange extract containing p-synephrine could significantly suppress appetite, increase energy, and decrease food intake without adverse effects [79].

Depression and anxiety are very often seen as a comorbidity in HD patients [80,81,82,83]. Kim et al. outlined the antidepressant-like activity in p-synephrine [84], and the effects of *Citrus aurantium* L. essential oil regarding the treatment of anxiety and depression has been discussed in several studies [85,86,87,88,89,90]. *Citrus aurantium* L. is an essential oil with a potential benefit for pain reduction and QoL improvement for HD patients. 

### 4.2. Phenols

Phenol is a hydroxybenzene, which is an aromatic organic compound. In pure form, it is a colorless-to-white crystalline [91].

The substances contained in WDT are gingerols [2,30], a group of volatile phenolic compounds [92,93]. Gingerols are categorized according to their alkyl chain length, e.g., (6)-gingerol, (8)-gingerol, and (10)-gingerol, where (6)-gingerol is the most abundant compound segregated from ginger, while other gingerols are present in lower concentrations [93,94]. Chemically, gingerols are methoxy-substituted phenols with an alkyl side-chain that carries one keto and one hydroxy group each [95]. 

Gingerols are known to have beneficial medicinal properties and exert remarkable pharmacological and physiological activities. Several pre-clinical studies have supported their role in treating several disorders such as diabetes, pain, fever, and inflammation [93,96].

A current hypothesis is that oxidative stress plays a major role in migraine pathogenesis [97,98,99]. Studies have demonstrated that gingerols possess a high antioxidant effect by inhibiting superoxide and nitric oxide production and suppressing lipid peroxidation [93,96,100,101,102,103,104,105,106]. It can be assumed that the established anti-oxidative activity of gingerols and specifically of (6)-gingerol might be advantageous for the treatment of HD [49,93,107].

A connection between prostaglandin and migraine-like attacks has already been shown [108,109]. Gingerols have a function as a suppressor of proinflammatory cytokines, inhibiting prostaglandin and leukotriene biosynthesis [110,111,112], and might therefore be a suitable antipyretic for prostaglandin-induced HD treatment [49,113].

Studies in human liver cell lines indicate that (6)-gingerol decreases inflammation and oxidative stress by decreasing mRNA levels of inflammatory factor interleukin 6 (IL-6), interleukin 8, and serum amyloid A1 [114]. In vivo models with treatment and topical application of (6)-gingerol to mice exhibited anti-inflammatory activity as well [115,116]. The reduction in inflammatory factors was shown to be an antidepressant effect in an animal model of depression [117,118]. Furthermore, gingerols play an important role in influencing the absorption of glucose by increasing cell surface distribution of the glucose transporter Type 4 (GLUT-4) protein [119,120], a glucose transporter that regulates the insulin-dependent glucose uptake in skeletal muscles [121,122]. Insulin attaches to the alpha-subunit of the insulin receptor and thus ensures an increase in phosphoinositide 3-kinase (PI3K. The signal transmission at the GLUT-4 caused by PI3K binding to the cell membrane results in an increase in glucose transport into the muscle cells [123,124].

A typical adverse effect of a hypoglycemic state is headache. The ICHD-3 classifies this HD as a headache attributed to other metabolic or system disorder ICHD-3 A10.8.2 [39]. A hypoglycemic state can be prevented by the increased absorption of glucose [125].

Many HD patients suffer from nausea and emesis before or during their migraine attacks [126,127]. Studies showed that gingerol could suppress emetic signal transmission in vagal afferent neurons by inhibiting the 5-hydroxytryptamine 3 as well as being antagonistic to acetylcholine receptors [128,129,130,131,132,133,134]. The efficacy of ginger compounds on the prevention of nausea and vomiting of various origins, albeit with the limits of the chemical stability of the gingerol compounds, was highlighted in a systematic review of randomized controlled trials [135]. Moreover, gingerols have been shown to selectively inhibit the inducible form of cyclooxygenase-2 (COX-2) but not the constitutive form cyclooxygenase-1 (COX-1). As inhibition of COX-1 is associated with gastrointestinal side effects; selective inhibition of COX-2 might help minimize these side effects [136]. Hence, using ginger to reduce nausea and emesis in HD patients would be a suggestive approach.

Gingerols have been shown to have further bioactive effects, such as being cardiotonic, antiemetic, anti-inflammatory, antitumor-promoting, anti-platelet aggregation in migraine [118], antifungal, analgesic, and antibacterial [125,137,138,139,140,141,142]. According to Zick et al., healthy humans can tolerate (6)-, (8)-, and (10)-gingerols till 2000 mg [143].

### 4.3. Isoflavonoids

Isoflavonoids are secondary metabolites of plants and a subclass of flavonoids [144]. They are defined by a B-ring attached at the C-3 position of their C-ring and are derived from the flavonoid biosynthesis pathway via liquiritigenin or naringenin [145]. The Isoflavonoid group includes isoflavans, isoflavonones, isoflavones, rotenoids coumestans, and pterocarpans. Isoflavonoid compounds have biological effects via the estrogen receptor. They are natural selective estrogen receptor modulators, have a structural similarity to 17-β-estradiol [146], bind especially to the β-estrogen receptor in the brain, have osteo-protective effects, and alleviate menopausal symptoms [147]. Migraine attacks are intensified due to increased estrogen levels in women [148]. WDT contains isoflavones [2,30] that could improve migraine caused by elevated estrogen levels and/or prevent menstrual-associated migraine [149,150]. 

Besides several flavonoids, WDT also contains isoflavonoids such as liquiritin and liquiritingenin [2,30]. Liquiritin is the 4’-O-glucoside of flavanone liquirtingenin [151]. Liquiritin and Liquirtingenin are characterized by their antioxidant, anti-inflammatory, anti-rheumatoidal, and neuroprotective effects [152,153,154].

### 4.4. Liquiritin, Liquiritingenin, and Isoliquiritigenin

*Radix Glycyrrhizae* (RG) is the rhizome of *Glycyrrhiza ruralness* Fisch., *Glycyrrhiza inflate* Bat., or *Glycyrrhiza glabra* L. from Leguminosae/Fabaceae, which are widely distributed in the northeast and northwest of China. The dried roots and rhizomes (GU), commonly known as licorice in Pharmacopeias [155,156], are one constituent of WDT. Licorice was shown to have antitussive, expectorant, and antipyretic effects and is mostly used for its therapeutic effects in alleviating cough, pharyngitis, bronchitis, and bronchial asthma [151,157,158].

The sweet-tasting bioactive saponin (Glycyrrhizin) is present in all Glycyrrhiza species. Glycyrrhizin can provoke hypertension, sodium salt and water retention, and potassium ion levels reduction [159,160]. Higher doses of glycyrrhizic acid (400 mg/day) have risky side effects, including cardiac dysfunction, edema, and hypertension [161]. Nevertheless, medicinal plants have beneficial chemical constituents, Liquiritin (LT), Liquiritingenin (LTG), and Isoliquiritigenin (ISL), that produce physiological changes of various health benefits [151,162]. There are at least 400 different chemical compounds in RG along with triterpenoid saponins, flavanones, chalcones, coumarins, and their glycosides [151,158,163,164]. 

The main bioactive flavonoid compounds in RG, LTG, and ISL were identified and isolated from the crude extract of *Glycyrrhiza uralensis* [165,166]. ISL demonstrates antioxidant, anti-inflammatory, antitumor, and hepatoprotective activities. Additionally, LTG is an estrogenic compound that acts as a selective agonist for the β-subtype estrogen receptor [151]. Moreover, derivatives of ISL and LTG were shown to have in vivo anti-diabetic activity [167].

LTG pre-treatment significantly reduced the LPS-induced depression symptoms in an animal model with a decrease in the levels of the proinflammatory cytokines in serum and hippocampus compared with the control LPS group [168]. LTG has been shown to lower the expression of brain-derived neurotrophic factor (BDNF) and p-TrkB (tropomyosin receptor kinase B) [169], indicating that the antidepressant and antianxiety activities of LG/LTG might be due to anti-inflammatory and BDNF/TrkB pathways [168]. In the hippocampus of the animal model of depression, LG upregulated the concentrations of 5-hydroxytryptamine (serotonin; 5-HT) and norepinephrine (NE) and acted on the PI3K/Akt/mTOR-mediated BDNF/TrkB pathway in the hippocampus [151,169].

These facts suggest that LTG could ease depressive-like symptoms in the mice model of depression [169]. Additionally, other reports indicated that LTG and ISL might act as major MAO inhibitors, which in turn are beneficial in the treatment of anxiety and depression and consequently as a preventive measure for HD [169,170,171].

In addition, ISL and LTG showed effective prevention of glutamate-induced toxicity by attenuation of mitochondrial malfunction and thus might help to inhibit neurodegeneration [172,173] and related HD symptoms. However, LTG showed no effect on capsaicin-induced activation of the transient receptor potential vanilloid-related (TRPV1 receptors). It concentration-dependently inhibited allyl isothiocyanate (AITC)-induced TRPA1 receptors activation. Additionally, LG treatment in HEK293 cells altered TRPM7-dependent inward/outward currents without alteration in cell proliferation and viability. Thus, hinting that its TRP channel-inhibiting properties may be of potential in developing novel and tolerable analgesic therapies [174]. 

Studies have shown that ISL could inhibit inflammation for parameters such as interleukin-1β (IL-1β) and tumor necrosis factor α (TNF-α), a signaling substance secreted by macrophages during inflammation [175,176,177,178]. Although the etiology of migraine is not fully understood, there is some evidence of increased levels of IL-1β and TNF-α in migraine and chronic tension-type headache [179,180,181]. The implementation of WDT containing isoflavonoid compounds might be suitable for patients in the treatment of HD. 

Liquiritin has been shown to inhibit progesterone metabolism by competitively inhibiting Aldo-keto reductase family 1 member C1, an enzyme transforming progesterone to an inactive form, thus limiting the biological effect of progesterone [182]. Considerable evidence allows a link between estrogen and progesterone and migraine [183,184,185,186]. Migraine appears more intermittently in adult women compared to men [187,188]. Menstrual-related migraine generally occurs at the time of menses in many migrainous women and exclusively with menses in some [189]. Menstrual-related migraine is often associated with other menstrual symptoms such as nausea, breast tenderness, and cramps. All symptoms appear to result from falling estrogen and progesterone levels [184]. As LG-induced inhibition of progesterone metabolism contributes to the stability of progesterone concentration levels, liquiritin might have a positive effect against migraine, nausea, and other menstrual-related symptoms. 

Chen et al. have described liquiritin’s antidepressant effects through fibroblast growth factor-2 enhancement by inhibiting neuroinflammation and maintaining synaptogenesis [190]. Treatment of mice with LTG followed by polysaccharides injection, which causes acute depressive behavior, has led to a decrease in proinflammatory cytokines IL-6 and TNF-α in serum and hippocampus when compared with the control [168]. These antidepressant and antianxiety activities of LTG can improve patients’ quality of life.

### 4.5. Triterpenes

Triterpenes are natural compounds derived biosynthetically from isoprene and consist of six isoprene units and 30 carbon atoms [191]. The vast majority of triterpenes are composed of tetra- or pentacyclic compounds [192]. Triterpenes form the basic structure of many saponins, tetrapenes, and steroids. They can be divided into three classes according to polarity: liphophilic triterpenes, highly oxidized triterpenes, and hydrophilic, glycosidic triterpenes [193]. 

Pachymic acid, a representative of the triterpenes, is contained in WDT as a bioactive component [2] that is found in the fungus *Poriae sclerotium* cocos albae (PC), a family of polyporaceae. It is claimed to have many pharmacological effects such as anti-inflammatory, antioxidant, antiemetic, diuretic, and antitumor [194,195,196].

Pachymic acid exerts an insulin-like, hypoglycemic activity by inducing glucose transporter type 4 gene expression and translocation to the plasma membrane in mammalian 3T3-L1 adipocytes, resulting in increased glucose uptake activity [197] as headache prevalence is greater in patients with diabetes than in non-diabetic patients [198]. 

Clinically, PC is mostly implemented in the treatment of hepatitis B, diabetes, cancer, metabolic syndrome, and modulation of the immune system [199,200].

### 4.6. Organic Acids

Organic acids are organic compounds with a carboxylic group as a functional group. They release hydrogen ions or hydronium ions in water and have the function of donating a proton [201].

WDT contains succinate as a bioactive compound [2]. Succinate, an important metabolite, is an esterification of succinic acid and interacts in various processes in the cell [202].

The most common curable nutrition disorder is iron deficiency. Iron deficiency anemia (IDA) in adults, mostly women, in industrialized countries has a prevalence of up to 5% [203,204]. The main symptoms of IDA are fatigue, vertigo, insomnia, depression, headache, brittle nails, etc. Serotonin, a neurotransmitter widely found in the central and peripheral nervous systems, plays a crucial role in migraine neurobiology [205]. An increase in 5-hydroxyindoleacetic acid in the urine during a migraine attack was observed by Gasparini et al. for the first time [206]. In migraine attacks, the level of serotonin is decreased in the central nervous system but increased in the peripheral nervous system [206]. As iron plays a primary role in the synthesis of serotonin, dopamine, and norepinephrine, IDA could be responsible for a reduced level of serotonin [207].

Studies have shown coherence between IDA, hemoglobin, and serum ferritin levels and headache and/or migraine occurrence, mainly in women [208,209]. There are several effective methods in the treatment of IDA. Supplement of iron protein succinylate should be considered in the therapy of HD caused by IDA [210] due to it being less adverse than iron infusion. 

In a randomized, double-blind, placebo-controlled clinical study of early menopausal women, succinate-based composition treatment of mice showed restoration of the estrous cycle and an increase in the weight and calcium content of bone tissue [211]. Additionally, in a randomized, placebo-controlled clinical trial in menopausal women, a succinate-based combination therapy substantially decreased most subjectively evaluated characteristics of menopausal syndrome and enhanced blood serum levels of estradiol fourfold. It has also relieved hot flushes and headache symptoms [211].

### 4.7. Polysaccharides

Polysaccharides, possessing a broad range of biological functions, are polymeric carbohydrates consisting of more than ten monosaccharide units, which are linked together by glycoside bonds [212]. Common examples of polysaccharides are glycogen, chitin, chitosan, starch, cellulose, agarose, and pectin [213]. 

WDT contains *Portae sclerotium Cocos algae* Polysaccharides (PCPs) as a bioactive component [2]. PCPs are the most abundant substances in *Poriae sclerotium cocos albae*, also known as Fu Ling in Chinese, which is an edible medical fungus [200]. Traditionally, Fu Ling has been used for medicinal purposes for more than a thousand years [214]. The pharmacological actions of PCPs in Fu Ling include antibacterial, antitumor, anti-hyperglycemic, immunomodulatory, anti-inflammatory, immunostimulatory, anti-oxidative, anti-aging, anti-hepatic, anti-diabetic, and anti-hemorrhagic fever effects [200,213,215,216]. Additionally, Sun et al. showed in their study that PCPs improved hyperglycemia, hyperlipidemia, and hepatic steatosis in mice [217].

PCPs find applications in cancer therapy and many other diseases. They have been described to reduce tumor growth and improve antioxidant enzyme activity when fed daily for 7 weeks in Wistar rats [218]. As PCPs have an antitumor cell proliferation effect and inhibit tumor growth [219,220], their medical administration might be beneficial to the treatment of secondary headache caused by one or more space-occupying intracranial tumors ICHD-3 7.4.1 [39]. 

The polysaccharide from the *Poria cocos* (accepted name: *Poriae sclerotium* cocos albae) was shown to improve hyperglycemia, hyperlipidemia, and hepatic steatosis via modulation of gut microbiota [217]. An enhancement in gut microbiota and a decline in inflammatory factors can have positive effects on improving gut and brain function. Additionally, it is suggested that probiotics might have a beneficial effect in reducing the frequency and severity of migraine attacks. It is worth mentioning that, similar to migraine, disorders of the brain involving depression and anxiety have been demonstrated to be associated with increased gut permeability [221], and probiotics boosting gut microbiota help to decrease gut permeability and prevent leaky gut syndrome [222].

In 2015, PCPs received approval from the Chinese Food and Drug Administration for treating various types of diseases such as cancer, hepatitis alone, or during chemoradiation therapy for cancer patients [213].

### 4.8. Alternative Therapies

Besides Chinese herbal medicine such as WDT, TCM also provides a broad spectrum of therapeutic modalities for HD such as acupuncture, moxibustion, cupping, tuina, exercises such as Tai Chi/Qi Gong, and dietary. However, the main focus of this study is the herbal medicine component of WDT.

#### 4.8.1. Acupuncture

The concept of acupuncture is principally to puncture an exact acupuncture point with an acupuncture needle, with or without manipulation after insertion (i.e., tonify, sedate). The goal is to achieve harmony in the movement of energy (Qi) in the meridians and bring balance in an imbalanced region, organ, or meridian [223].

Although the exact mechanism of acupuncture is difficult to research and not yet completely understood, clinical trials have shown good evidence of acupuncture efficacy in the treatment of HD and for the prevention of migraine, TTH, or chronic HD [224]. As adverse events are mild and serious adverse events are rare, acupuncture can be considered a safe treatment [225].

#### 4.8.2. Topical Medicines

The skin is the largest organ of the body. Any abnormalities in the body can manifest on the skin surface. People suffering from HD, especially migraine, tend to experience cutaneous allodynia (CA) [226]. The International Association for the Study of Pain defines allodynia as “pain due to a stimulus that does not normally provoke pain”. Topical medicines in various forms such as ointments, oils, pastes, creams, lotions, foams, gels, tincture, powders, spray, and patches can be applied either to the skin or to mucous membranes. This may be used topically to treat pain or other ailments in a particular region of the body [227]. These applications may also be effective in treating CA and/or HD. St. Cyr et al. could show in their study that STOPAIN 6% menthol gel was safe and significantly reduced headache intensity by two hours after the gel application [228]. Yuan R et al. revealed in their review that aromatic plant essential oils had pain-relieving effects in migraine patients due to the suppression of neurogenic inflammation and pain sensitization [229]. However, there are limitations to skin therapies in case of skin injuries and/or infection. Moreover, the local application of the skin cannot reach a distant area of pain. Therefore, systemic application of a herbal TCM formula such as WDT by ingestion can be a better choice in the case of HD [230]. 

### 4.9. Western Medicine

The Western medicine pharmacotherapy treatment for HD includes acute and preventive medications [42]. The gold standard of migraine and other related HD suggest the usage of triptans, non-steroidal anti-inflammatory drugs (NSAIDs), and analgesics. The prophylactic treatment contains antidepressants, anticonvulsants, β-blockers, antihypertensive medications, CGRP-Inhibitor, calcium channel antagonists, magnesium, coenzyme Q10, riboflavin, and botulinum toxin type A [231]. Analgesics and NSAIDs are the standard therapy for acute TTH [232].

If patients repeatedly consume acute headache medication too often, this can result in a medication overuse headache [233]. 

## 5. Conclusions

Research regarding bioactive compounds of WDT is very rare. Based on our research information, only Zhang et al. [2] and Wu et al. [30] have examined the bioactive compounds of WDT in their studies. To the best of our knowledge, no clinical trial has been completed regarding the treatment of any type of headache with WDT. Therefore, this study aimed to explore the pharmacological reaction of the API containing WDT associated with HD.

The current study gathers information on WDT compounds, which can be used as a guideline for physicians to help HD patients and to improve their QoL. The alkaloid p-synephrine can reduce primary and secondary headaches caused by rhinitis patients through its vasoconstrictive characteristics. By increasing the fat-burning rate and the expected weight loss, p-synephrine could also contribute to life quality improvement of obese patients and related migraine headaches. Gingerols have a promising positive effect in reducing migraine through their anti-oxidative effect and an inhibitory effect on controlling prostaglandin-biosynthesis-induced migraine through the inhibition of prostaglandin biosynthesis and in working against hypoglycemia-caused migraine by increasing glucose transport into muscle cells. Isoflavones can contribute to the control of migraine caused by increased estrogen levels as well as menstrual-associated migraine. The isoflavonoids liquiritin and liquiritingenin might exert a positive effect on Interleukin 1β-induced headache. Liquiritin shows potency to have a beneficial effect on menstrual-related migraine and nausea by inhibiting progesterone metabolism. Liquiritigenin’s antidepressant and anxiolytic effects are expected to positively contribute to life quality improvement in HD patients. Hypoglycemic activity of pachymic acid exerts a positive effect against diabetes-caused secondary headache, whereas succinate-based composition helps control general menopausal and specifically headache symptoms by restoring the estrous cycle. Polysaccharide PCPs can contribute to migraine control by their anti-oxidative, anti-inflammatory, and antitumor effects and as probiotics enhancing gut microbiota composition.

These findings suggest that WDT, as a combination of these APIs, might be beneficial in treating several diseases, particularly for patients with phlegm turbidity HD. Currently, there is no sufficient clinical research investigating the pharmacological effects and the mechanisms of action of WDT. Consequently, more clinical trials are needed to better understand the treatment of HD with WDT.

High-quality clinical research over a long period with a larger patient group should be carried out to evaluate the therapeutic effectiveness of WDT in HD to investigate if WDT could be a suitable prescription for people with HD, especially migraine and TTH.

Our study lacks all the other clinical trials completed for various diseases other than HD. Moreover, the effect sizes of the API in WDT were not examined to investigate the etiopathogenetic mechanism of WDT. Nevertheless, our article contains HD as a primary and secondary disorder to other pathologies. Thus, it brings a broader spectrum of herbal use for therapeutic and prophylactic treatment for all types of HD.

This manuscript is significant because it holds collective evidence to address the multipotency of WDT and their bioactive components in detail. In conclusion, this study raises the importance of a conclusive clinical trial of WDT with large samples for patients treating migraine and TTH.

## Figures and Tables

**Table 1 medicines-09-00022-t001:** WDT herbal components species and function.

Species	Function	Reference
*Pinelliae Rhizoma* (*Pinellia ternata* (Thunb.) ^1^ Makino)	Ceases cough, dissolves phlegm, dries dampness, stops vomiting, possesses antitumor effects	[3]
*Caulis Bambusae* In *Taenia* sp. (*Bambusa tuldoides Munro*) ^1^	Arrests vomiting, alleviates fever, abdominal pain, diarrhea, chest diaphragm inflammation, has antifatigue attributes, regulates hypertension and hyperlipidemia, reduces aggravation	[4,5]
*Fructus Aurantii Immaturus* (*Citrus aurantium* L.) ^1^	Helps gastrointestinal disorders, is anti-coagulation, eliminates food stagnation by guiding the Qi downwards, has antianxiety properties	[6,7]
*Citri reticulate* Pericarpium (*Citrus reticulata* Blanco) ^1^	Dissolves phlegm, dries dampness, promotes Qi, strengthens spleen, has antiasthmatic characteristic	[8,9,10]
*Glycyrrhizae Radix* et Rhizoma (*Glycyrrhiza uralensis* Fisch) ^1^	Tonifies Qi and the spleen, harmonizes the action of all herbs in a prescription, and eliminates the toxicity of herbs	[11,12]
*Poria Cocos* (*Poriae sclerotium* cocos albae) ^1^	Strengthens the spleen and harmonizes the stomach, has antianxiety properties, is calmative, has a soothing diuretic effect	[13,14]
*Zingiberis Rhizoma* (*Zingiber officinale* Roscoe) ^1^	Has an antiemetic effect, alleviates pain, harmonizes the stomach and spleen, warms the core and the lungs, removes cold	[15,16]
*Jujubae Fructus* (*Ziziphus jujuba Mil*) ^1^	Nurtures the blood, has a calmative effect, promotes Qi, tonifies the stomach and spleen, regulates digestive system, reduces the toxicity of herbs	[17,18]

WDT origins and functions listed in Table 1**.**
^1^ The accepted Nomenclatural name of the species validated by www.theplantlist.org (accessed on 19 May 2021).

**Table 2 medicines-09-00022-t002:** Application of WDT in various syndromes.

References	Title	Syndrome
[24]	Treatment of Insomnia with Traditional Chinese Herbal Medicine.	Insomnia
[27]	Wendan decoction for primary insomnia.	
[30]	Wen-Dan Decoction Improves Negative Emotions in Sleep-Deprived Rats by Regulating Orexin-A and Leptin Expression.	Negative Emotions
[26]	Consistent Efficacy of Wendan Decoction for the Treatment of Digestive Reflux Disorders.	Digestive disorder
[31]	Wendan decoction for dyslipidaemia: Protocol for a systematic review and meta-analysis.	Dyslipidaemia
[23]	Metabolomic investigation into molecular mechanisms of a clinical herb prescription against metabolic syndrome by a systematic approach.	Metabolic syndrome
[25]	Efficacy of the wen dan decoction, a Chinese herbal formula, for metabolic syndrome.	
		Psychiatric disorders
[22]	Behavioural screening of zebrafish using neuroactive traditional Chinese medicine prescriptions and biological targets.	Major depressive disorder
[19]	Wendan decoction (Traditional Chinese medicine) for schizophrenia.	Schizophrenia
[21]	Effects of Wen Dan Tang on insomnia-related anxiety and levels of the brain-gut peptide Ghrelin.	Anxiety
[32]	Wen Dan Decoction for haemorrhagic stroke and ischemic stroke.	Stroke
[28]	Systems Pharmacology Dissection of Traditional Chinese Medicine Wen-Dan Decoction for Treatment of Cardiovascular Diseases.	Cardiovascular Diseases

Application of WDT in various syndromes listed in Table 2.

**Table 3 medicines-09-00022-t003:** A list of the bioactive compounds contained in WDT, as defined by Zhang et al. [2].

No.	Compound	PubChem CID ^1^	Chemical Structure ^1^
1	Synephrine		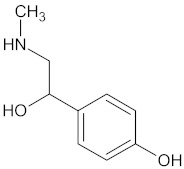
2	Succinate	160419	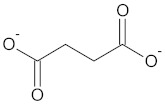
3	Liquiritin	503737	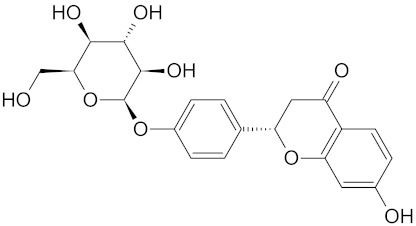
4	Eriocitrin	83489	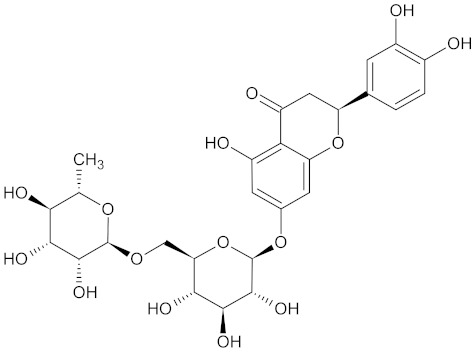
5	Rutin	5280805	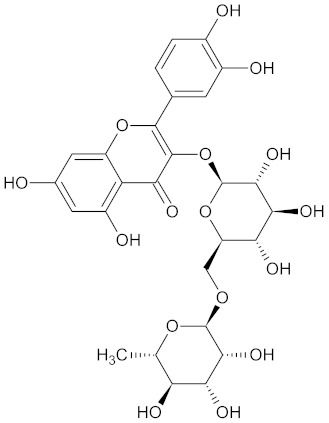
6	Narirutin	442431	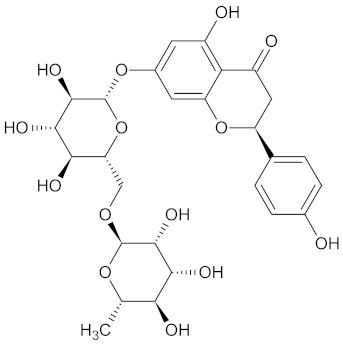
7	Naringin	442428	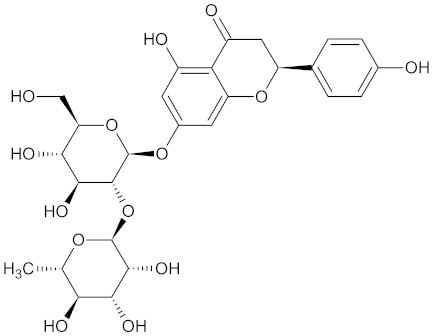
8	Hesperidin	10621	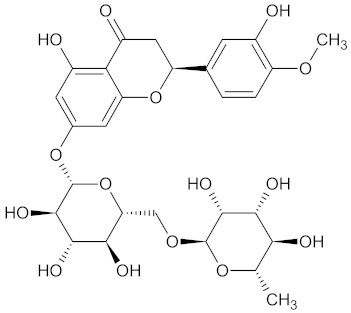
9	Neohesperidin	442439	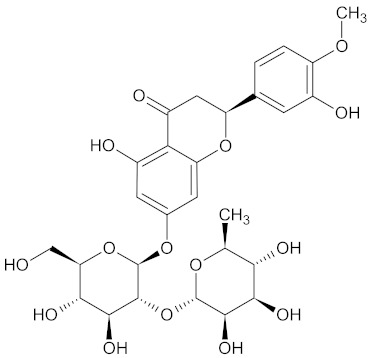
10	Liquiritigenin	114829	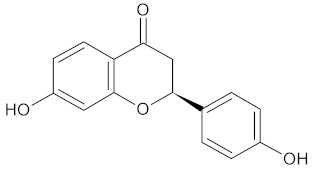
11	Isoliquiritin	5318591	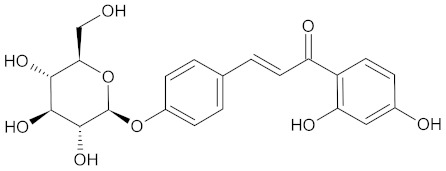
12	Didymin	16760075	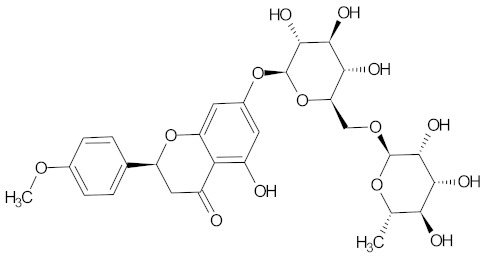
13	Poncirin	442456	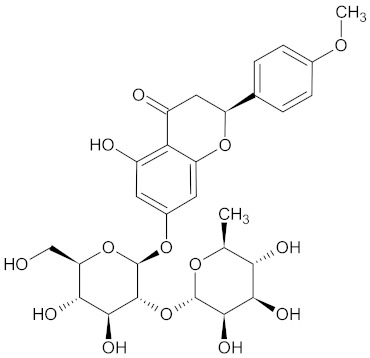
14	6-Gingerol	442793	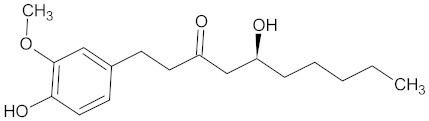
15	Tangeretin	68077	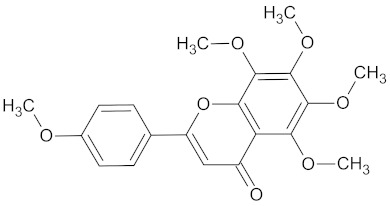
16	8-Gingerol	168114	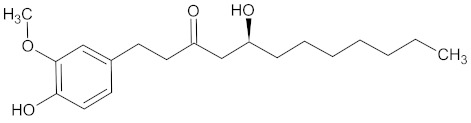
17	10-Gingerol	168115	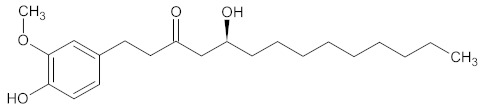
18	Pachymic acid	5484385	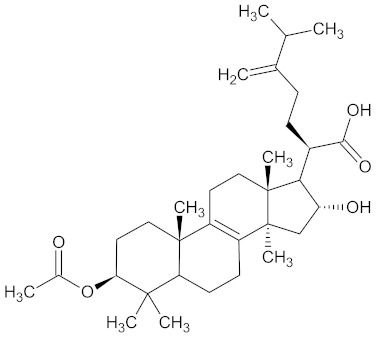
19	Dehydropachymic acid	15226717	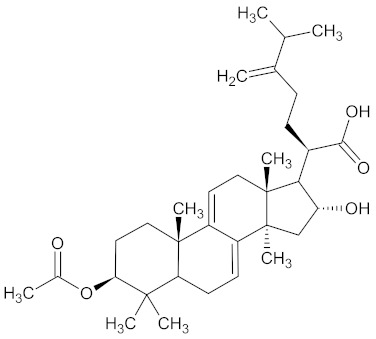

A list of the bioactive compounds contained in WDT, as defined by Zhang et al., is listed in Table 3 [2]. ^1^ All chemical structures were redrawn with (ACD/ChemSketch Freeware) [53] after being retrieved from National Centre for Biotechnology Information (2020). PubChem Compound Summary for CID number above from https://pubchem.ncbi.nlm.nih.gov/compound (Retrieved 25 September 2020).

## Data Availability

All data generated or analyzed during this study are included in this published article.

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
