# Peer review of "Wen Dan Tang: A Potential Jing Fang Decoction for Headache Disorders?"

_medicines, 2022, doi:10.3390/medicines9030022_

Round 1
Reviewer 1 Report
This manuscript summarizes the pharmacological mechanism of some active ingredients in WDT in the treatment of headache, which has certain enlightening significance for the study of WDT. However, there are some details in the article that need to be added or revised:
- According to the theory of traditional Chinese medicine, the prescription be consistenting with the syndrome type is the precondition for the prescription to exert its curative effect. For WDT, the type of headache it treats should be consistent with phlegm turbidity. Therefore, this point should be clearly stated in the manuscript, so as not to lead to misunderstanding that any headache WDT can be treated.
- The clinical evidence of WDT in the treatment of headache was inquired in the English literature database, but no corresponding results were found. The main reason is that the clinical application of WDT is mainly in China, limited by language and the differences between Chinese medicine and Western medicine, articles are mainly published in Chinese. If the Chinese literature database (such as http://www.cnki.net) is consulted, there will be many clinical reports on WDT for headache. However, high-quality, multicenter, and large-scale evidence-based medical evidence for WDT in the treatment of headache is still lacking.
- “Zingiber officinale Roscoe (Gan Jiang)” should be changed to “Zingiber officinale Roscoe (Sheng Jiang)”.
- “Sun Si Mao” should be changed to “Sun Si Miao”.
- “Prescriptions Worth a Thousand Pieces of Gold for Emergencies” should be translated as “Essential Recipes for Emergent Use Worth A Thousand Gold”.
- The WDT, which contains eight Chinese medicines, should be derived from the book “Treatise on Three Categones of Pathogenic Factors” exactly. However, the WDT in the book “Essential Recipes for Emergent Use Worth A Thousand Gold” has no Poriae sclerotium cocos albae (Fu Ling) and Ziziphus jujuba Mil (Da Zao).
- In “4 Discussion”, the pharmacological effects of each active ingredient were mainly discussed and analyzed. so the title 5-11 should be placed under the title “4 Discussion”, and the title number of “12 Conclusion” should be changed to “5 Conclusion”.
Author Response
Dear reviewer,
We thank you for your comments and your valuable recommendations, we really appreciate your advice. We revised our review and integrated your recommendations and have made the improvement changes accordingly.
Your comment 1:
For WDT, the type of headache it treats should be consistent with phlegm turbidity.
In conclusion we have changed it into: These findings provide some hints that WDT as a combination of these API might be beneficial in the treatment of several diseases, particularly, for patients with phlegm turbidity HD.
Your comment 2:
If the Chinese literature database (such as http://www.cnki.net) is consulted, there will be many clinical reports on WDT for headache. However, high-quality, multicenter, and large-scale evidence-based medical evidence for WDT in the treatment of headache is still lacking.
We are aware that articles regarding WDT are mainly published in Chinese. Since access located in Switzerland, Chinese databases such as the Chinese National Knowledge Infrastructure (CNKI), China Science and Technology Journal Database (VIP), and Wanfang database could not be accessed from Swiss internet IP address. Reviews and research articles published in these mentioned databases unfortunately were not considered.
Your comment 3:
“Zingiber officinale Roscoe (Gan Jiang)” should be changed to “Zingiber officinale Roscoe (Sheng Jiang)”.
Zingiber officinale Roscoe (Gan Jiang) has been changed into Zingiber officinale Roscoe (Sheng Jiang).
Your comment 4:
“Sun Si Mao” should be changed to “Sun Si Miao”.
Sun Si Mao has been changed to Sun Si Miao.
Your comment 5:
“Prescriptions Worth a Thousand Pieces of Gold for Emergencies” should be translated as “Essential Recipes for Emergent Use Worth A Thousand Gold”.
Prescriptions Worth a Thousand Pieces of Gold for Emergencies has been translated as Essential Recipes for Emergent Use Worth A Thousand Gold.
Your comment 6:
The WDT, which contains eight Chinese medicines, should be derived from the book “Treatise on Three Categones of Pathogenic Factors” exactly. However, the WDT in the book “Essential Recipes for Emergent Use Worth A Thousand Gold” has no Poriae sclerotium cocos albae (Fu Ling) and Ziziphus jujuba Mil (Da Zao).
We have changed it into: Wen Dan Tang (WDT) also known as “Warm the Gallbladder decoction” is a classical famous Chinese herbal formula containing eight components derived from the book Treatise on Three Categories of Pathogenic Factors:
Your comment 7:
In “4 Discussion”, the pharmacological effects of each active ingredient were mainly discussed and analyzed. so the title 5-11 should be placed under the title “4 Discussion”, and the title number of “12 Conclusion” should be changed to “5 Conclusion”.
Title 5-11 has been placed under the title 4 Discussion, and the title number of “12 Conclusion” has been changed to “5 Conclusion”.
Reviewer 2 Report
Pradhan et al submit their manuscript on reviewing the use of Wen Dan Tang (WDT) decoction and its derivatives in headache disorder. In the Chinese medicine field, it is important, from time to time, to review traditional Chinese medicine decoction in a modern scientific way. However, one of the major concerns in this manuscript is about the efficacy of using the WDT in treating headache disorder. As mentioned by the authors, even they cannot find a study on WDT in headache disorder. Then they suggest some of the active components derived from WDT and try to relate their beneficial effect in some of the risk factors in headache such as neurovascular or neuropsycharitic problems. Nevertheless, the WDT decoction doesn't need to have a beneficial effect on headaches. Authors may change their focus of the current manuscript such as the effect of WDT, their derivatives, or active components in neurovascular or cardiovascular disorder.
There are a few minor points that can be modulated:
- The authors mentioned the extraction methods. It will be better to compare different extraction methods, i.e. using different extraction solvents, to characterize some of the pharmacological aspects such as toxicity, as well as the chemical profiles.
- Authors may compare the benefits and the adverse effect of WDT with the currently used medicine in the disease model they want to discuss
Author Response
Dear reviewer,
We thank you for your comments and your valuable recommendations, we really appreciate your advice. We rewrote our review and integrated your recommendations and have made the improvement changes accordingly.
Your comment 1:
- The authors mentioned the extraction methods. It will be better to compare different extraction methods, i.e., using different extraction solvents, to characterize some of the pharmacological aspects such as toxicity, as well as the chemical profiles.
We have added two more extraction methods.
Traditionally, the compounds of WDT are immersed in 1200 mL water for 30 minutes and then boiled in a decoction pot. After the boiling process, the decoction simmers for 30 to 40 minutes. The extract is filtered, and usually, the dosage of 300 mL/day is administrated for drinking. Zhang et al. stated in their study that the extraction of WDT applying the UHPLC-QQQ-MS/MS method was more beneficial than the traditional extraction technique with water because the boiling process negatively affects the stability of the molecule gingerol. In addition, instead of methanol, acetonitrile was utilized for the chromatography, and thereby better isolation of the individual components could be achieved. The complete extraction method could be performed within 30 minutes in the lab. This procedure is thus not more time-consuming than the traditional way and purer gingerols extract can be obtained with it [2].
Your comment 2:
- Authors may compare the benefits and the adverse effect of WDT with the currently used medicine in the disease model they want to discuss.
We have provided alternative therapies which are currently used medicine in HD and stated the safety of WDT.
Alternative therapies consist of acupuncture, topical medicines, and Western medicine.
WDT is considered relatively safe, economical, and obtainable [19].
As for the adverse effect of WDT we have added: The most common adverse events reported by Huang et al. in their meta-analysis were loose stool, dizziness, poor appetite, fatigue, xerostomia. However, the observed AEs were mild [25].
Reviewer 3 Report
Firstly, the authors aimed to find evidence-based clinical trails of WDT in the treatment of migraine and TTH by searching online databases including PubMed, Medline, Cochrane Library, AcuTrials, Embase, Semantic Scholar, Jstor, internet research and review of ancient and modern Chinese medical textbooks. However, some papers published in Chinese have reported the positive effects of WDT for headache disorders. Secondly, some references were misquoted and misapplied. For instance, Ref 6 and Ref 12 were basic researched and could not directly support clinical usage of WDT. Thirdly, the majority of results and studies for the single compounds in WDT were irrelevant to headache disorders. More importantly, the single compound could not represent single herb even WDT. Finally, some basic mistakes existed. For example, glycyrrhizic acid (GA) and 18β-glycyrrhetinic acid (18βGA) are saponins instead of isoflavonoids.
Author Response
Dear reviewer,
We thank you for your comments and your valuable recommendations, we really appreciate your advice. We revised our review and integrated your recommendations and have made the improvement changes accordingly.
Your comment 1:
- However, some papers published in Chinese have reported the positive effects of WDT for headache disorders.
We are aware that articles regarding WDT are mainly published in Chinese. Since access located in Switzerland, Chinese databases such as the Chinese National Knowledge Infrastructure (CNKI), China Science and Technology Journal Database (VIP), and Wanfang database could not be accessed from Swiss internet IP address due to technical issues. Reviews and research articles published in these mentioned databases were unfortunately not considered.
Your comment 2:
- Secondly, some references were misquoted and misapplied. For instance, Ref 6 and Ref 12 were basic researched and could not directly support clinical usage of WDT.
We deleted Ref 6 and Ref 12 from the following text: Additionally, there have been numerous research showing WDT usage in the treatment of various diseases that might lead to HD [19–21, 23–27, 29]. Before the references were [3–13].
- Thirdly, the majority of results and studies for the single compounds in WDT were irrelevant to headache disorders.
In fact, we could show that individual ingredients of WDT have a beneficial effect on headache disorders and its complication like nausea and emesis before or during migraine attacks.
Many HD patients suffer from nausea and emesis before or during their migraine attacks [126,127]. Studies showed that gingerol could suppress emetic signal transmission in vagal afferent neurons by inhibiting the 5-hydroxytryptamine 3 as well as being antagonistic to acetylcholine receptors [128–134]. The efficacy of ginger compounds on the prevention of nausea and vomiting of various origins, albeit with the limits of the chemical stability of the gingerol compounds was highlighted in a systematic review of randomized controlled trials [135]. Moreover, gingerols have been shown to selectively inhibit the inducible form of cyclooxygenase-2 (COX-2), but not the constitutive form cyclooxygenase-1 (COX-1). As inhibition of COX-1 is associated with gastrointestinal side effects, selective inhibition of COX-2 might help minimization of these side effects [136]. Hence using ginger to reduce nausea and emesis in HD patients would be a suggestive approach.
- More importantly, the single compound could not represent single herb even WDT.
Yes this is true, therefore the different components of WDT have beneficial effect that is summed up to the total positive effect in treatment of headache disorders.
This study suggests that the bioactive compounds found in WDT composition show potential in the treatment of patients with neurological, psychiatric disorders, cardiovascular diseases, metabolic syndrome, and digestive disorders. Some coherence between WDT in headache reduction and the improvement of the quality of life in migraine and TTH patients could be evaluated, showing positive results of WDT for these patients with migraine and TTH.
- Finally, some basic mistakes existed. For example, glycyrrhizic acid (GA) and 18β-glycyrrhetinic acid (18βGA) are saponins instead of isoflavonoids.
We have modified the sentence into:
Saponins like Glycyrrhizic acid (GA) and 18β-glycyrrhetinic acid (18βGA) are active metabolites of GU extract that mimic aldosterone action in human physiology.
Round 2
Reviewer 2 Report
The revised manuscript has been improved and the authors have clarified some of the concerns. The manuscript requires a few English proofread.
Author Response
Dear reviewer,
We appreciate your comments. The manuscript has been English proofread and edited by MDPI.
Reviewer 3 Report
Some pharmacological and activity results and studies of the single compounds in WDT showing weak/no direct relationship with headache disorders could be eliminated. The sentences about saponins under the subtitle "4.4. Liquiritin, Liquiritingenin and Isoliquiritigenin" are improper and shall be deleted.
Author Response
Dear reviewer,
We appreciate your comments. We have removed the sentences about saponins as you recommend. The pharmacological effect of a single compound in WDT showed the indirect effect of headache disorders as written in case of pachymic acid that has hypoglycemic activity increasing glucose translocation inside the cell which work positively on headache prevalence in diabetic patients subtitle "4.5. Triterpenes".